# Envisioning Happy Places for All: A Systematic Review of the Impact of Transformations in the Urban Environment on the Wellbeing of Vulnerable Groups

**Marica Cassarino** [1,*], **Sina Shahab** [2] and **Sara Biscaya** [3]

1    School of Applied Psychology, University College Cork, T23 TK30 Cork, Ireland
2    School of Geography and Planning, Cardiff University, Cardiff CF10 3WA, UK; ShahabS@cardiff.ac.uk
3    School of Science, Engineering & Environment, University of Salford, Manchester M5 4WT, UK;
     s.biscaya@salford.ac.uk
*    Correspondence: mcassarino@ucc.ie

**Abstract:** Urban planning and design can impact mental health, but it is unclear how ever-growing and changing cities can sustain the psychological wellbeing of vulnerable groups, who are among the most mentally sensitive to spatial inequalities. This systematic review synthesised quantitative and qualitative studies on urban design interventions and their impact on wellbeing in vulnerable groups. Using the Preferred Reporting Items for Systematic Reviews and Meta-analyses (PRISMA) guidelines, we searched five online databases from inception to May 2020. A total of 10 papers were included. We found mixed evidence of benefits for wellbeing linked to urban regeneration projects or focused interventions (green spaces, transport, security). Interventions that were centred around participation, sustainable living, and quality of design (e.g., perceived sense of safety) were associated with increased residential satisfaction and wellbeing, particularly among low-income communities and women. Risk of bias was low to medium, but there was high methodological heterogeneity; studies were mainly from Western countries, and none of the included studies investigated the experiences of people with disabilities, migrants, or racial minorities. This review highlights the importance of inclusive and sustainable design interventions to create happy places for all strata of society, although further investigation is warranted.

**Keywords:** wellbeing; vulnerable groups; equitable urban design; urban intervention; transformative urban development

## 1. Introduction

In an increasingly urbanised society, and particularly in light of the recent global pandemic, there is an acknowledgment that the design of urban spaces can have a considerable impact on health and wellbeing inequities [1–3]. Developing sustainable cities and communities (SDG11), promoting health and wellbeing (SDG3), and reducing inequalities (SDG10) are among the 17 ambitious Sustainable Development Goals (SDGs) set by the United Nations [4]. To achieve these goals, there is a growing need to better understand how urban planning and design can support and sustain the psychological wellbeing, or happiness, of all citizens in an equitable manner [5,6]. Indeed, the Global Happiness Policy Report 2018 [7] considers urban planning as one of the key determinants of happiness. In this review, in line with Dodge et al. [8], we consider happiness in the broad sense of a condition of positive and sustained subjective wellbeing [9] encompassing both the hedonic dimension of positive affective states [10] and the eudaimonic dimension of positive psychological functioning, flourishing and development (see for a discussion [11]). To this end, and in line with previous research on the topic [11,12], the terms happiness and psychological wellbeing are used interchangeably hereafter.

Frameworks of urban "liveability" increasingly suggest that the design of urban outdoor spaces can be a determinant of health inequalities [13], but the main focus of research

has been either on physical health [14] or, on the other hand, on the social determinants of mental health [15–17]. While environmental psychology has stimulated research on the psychological influence of physical spaces [5,18], happiness, or subjective wellbeing, has only recently started to receive appropriate attention in relation to urban form and sustainable urban development [16]. Since the publication of Montgomery's *Happy City* book [19], there have been increasing attempts to develop ways to measure the link between urban development and happiness [12]; however, there is mixed evidence on the association between urban transformation and psychological wellbeing. For instance, a study in Spain found that urban renewal projects focused on active travel and inclusive public spaces benefited residents' mental wellbeing [20], whereas another study in the UK observed no sizeable psychological benefits associated with housing regeneration projects [21]. Similarly, a recent systematic review of the impact of urban interventions on mental health [22], found limited and weak evidence for a positive impact of improvements to green infrastructure on quality of life and social isolation.

Therefore, more evidence is required to establish what urban design factors contribute to making cities not only "healthy" but also "happy places", i.e., promoting and sustaining psychological wellbeing. To this end, the Green-Active-Prosocial-Safe Places (GAPS) Framework [23] suggests that urban spaces can contribute to positive psychological outcomes if they enable access to nature, offer opportunities for action and social interactions, and afford accessibility through a sense of safety. Thus, urban spaces that are created to foster a sense agency (or a positive eudaimonic state) and belonging (i.e., positive hedonic or affective state) may promote happy citizens [24]. Notwithstanding, city design can influence wellbeing across all strata of society, there is growing recognition that vulnerable groups [25] are among the most mentally sensitive to spatial inequities [23] because they are susceptible to physical enablers or barriers for wellbeing present in their communities [26]. While a unified definition of "vulnerable groups" has not been achieved globally [25], there is a generic agreement that this denomination includes those that experience a higher risk of poverty and social exclusion than the general population. From an urban design perspective, groups including older people, families with children, women, economically disadvantaged groups and minorities can be considered vulnerable to health and social risks determined by the physical environment they inhabit [23,26]; thus, they are those who may benefit the most from urban transformations that promote psychological health [23]. Indeed, holistic views of happiness or psychological wellbeing (see for a discussion [8]) describe it as a complex dynamic state between the individual's resources/vulnerabilities and the opportunities/challenges posed by the environment; the more opportunities the environment offers, the more the benefits, particularly for those with limited physical, mental or social resources. Applied to urban design, this view on psychological wellbeing suggests that equity of urban spaces is crucial to create "happy places", i.e., places that support the psychological needs of those individuals who are the most vulnerable; this has been advocated, for instance, within global initiatives around Child Friendly [27] or Age Friendly Cities [28], or Universal Design [29]. While previous systematic reviews have established the evidence base for the impact of urban design on mental health [22], it is unclear what urban factors contribute to create equitable "Happy Places for All"; which is the focus of the present systematic review.

In trying to answer the question on the urban factors that promote the creation of "Happy Places for All", this systematic review aimed to address some important evidence gaps. Firstly, to the best of our knowledge, this is the first review to focus on the psychological impact of changes in urban planning and design, whereas most investigations of environmental determinants of psychological wellbeing have focused mainly on the status quo, i.e., comparing different local communities with/without certain characteristics [30,31]. While such evidence is pivotal to develop an understanding of urban inequalities, a synthesis of the literature pertaining the role of urban interventions for psychological wellbeing is needed to identify equitable strategies for urban change.

Secondly, while previous reviews focused on controlled studies to evaluate the objective effectiveness of urban interventions [22], this systematic review included different types of study designs to gain insights on how vulnerable groups experience urban changes subjectively, so to enrich our understanding of what urban regeneration might entail for them [20]. Participatory studies with low-income groups or children [32,33], for instance, have highlighted the value of using users' 'bottom-up' expertise, especially with vulnerable groups who have first-hand experience of spatial inequities, to develop more effective and empowering urban living strategies [34]. To this end, we set out to explore subjective experiences both through an evaluative lens (i.e., how people feel about change that has occurred) and in terms of the level of participation and empowerment afforded to vulnerable groups throughout the transformation process [32–34].

Lastly, researchers in different fields have highlighted the need to develop transdisciplinary perspectives on urban development in order to achieve a more comprehensive understanding of the complex associations between urban form and health/wellbeing [22,35,36]. This systematic review drew from expertise in architecture, urban planning, and psychology fields, to assess the evidence through a transdisciplinary lens and to critically compare the effectiveness of comprehensive and focused urban interventions, in order to clarify the larger benefits for mental wellbeing.

Thus, this systematic review had two main objectives:

(1) To clarify whether and how urban planning and design interventions impact the psychological wellbeing of vulnerable groups;
(2) To investigate vulnerable groups' experiences and perceptions of the process of transformation in the urban environment, particularly with regards to the experience of participation and involvement.

## 2. Materials and Methods

### 2.1. Protocol and Registration

This systematic review was conducted in line with the preferred reporting items for systematic reviews and meta-analyses (PRISMA) guidelines [37], and the PRISMA checklist can be found in Supplementary File S1. The protocol for this systematic review is registered with PROSPERO (record # CRD42020182778).

### 2.2. Eligibility Criteria

We adopted the following population, interventions, comparators, outcomes, and study designs (PICOS) criteria to select studies (see Supplementary File S2 for a detailed table of inclusion and exclusion criteria):

- *Population*: Vulnerable groups, based on United Nation criteria, which included children, women, older people, people with functional limitations, people with low socioeconomic status, racial minority groups, and migrants. These categories did not form part of the search string, but were used during study selection;
- *Intervention* (exposure): Studies were included if they (a) described one or more urban intervention (i.e., change in planning or design of the urban environment as opposed to describing the status quo of urban design), and (b) were based in urban areas as opposed to rural, natural, or wild areas;
- *Comparison*: It was anticipated that there would be limited availability of controlled studies in this area. However, should controlled studies be found, we would use urban areas or groups where the intervention had not been carried out as a comparator
- *Outcome*: Psychological wellbeing or any related psychological status. Given the interchangeability with which happiness and psychological wellbeing are used in the literature, as discussed in the introduction, we adopted here a broad definition [8] to encompass both direct and indirect measures of "being psychologically well" in the community. In particularly, we considered as eligible any studies that provided a measure related to sense of belonging or agency in the community.

- *Study Design*: Observational studies (quantitative or qualitative), controlled studies, case studies.

   In line with PICOS guidelines, all the above five criteria needed to be met for a study to be included in the review. Further details on the logic used to select studies are presented in Supplementary File S2.

### 2.3. Search

   The search string and terms can be seen in Supplementary File S2. We searched Web of Science, Scopus, EBSCO Academic Search Complete, PsychInfo and PubMed, from inception until May 2020. We limited the search to records in the English language and empirical publication types. The reference lists of included studies were also hand searched. Where the full-text of an abstract could not be accessed by the research team, an attempt was made to contact the study authors via email. All results were imported into the Rayyan citation management software [38], where duplicate citations were screened and removed.

### 2.4. Study Selection and Data Collection

   We screened records employing a two-stage process. In the first stage, titles and abstracts were screened independently by the research team (SB, MC, and SS). The records were divided into three groups, and a pair of the three authors reviewed each group within the Rayyan interface. In the second stage, the selected full-texts were divided once again into three groups and each group was screened by a pair of the authors to confirm inclusion in the final review. At each stage, any conflicts between two authors were moderated by a third author. Before initiating the screening process, the inclusion and exclusion criteria were piloted by the research team in a sub-sample of 30 records, with each pair of authors screening 10 records to ensure that the same criteria were being used. The piloting stage and involvement of all authors in the screening process were adopted to limit potential bias or error in study selection.

   We extracted data from the included studies using an ad hoc data extraction form created by the research team. All extracted data, which related to study reference, methods, and results, is included in Supplementary File S3. All three authors contributed to data extraction and disagreements were resolved by consensus and moderation.

### 2.5. Risk of Bias of Individual Studies

   We employed the mixed methods appraisal tool (MMAT) [39] to assess the quality of the studies included in the review (see Supplementary File S4). Each study was identified and assessed based on its methodology. The risk of bias was assessed by the two authors independently (MC and SS), and any discrepancies were moderated by a third author (SB).

### 2.6. Synthesis of Results

   A meta-analysis was not possible due to heterogeneity of populations, interventions, or outcomes in the included studies. The main characteristics of the included studies are described based on setting, methodology, populations, outcomes, and approaches. A detailed narrative synthesis is provided based on the approach of the intervention (comprehensive vs. specific). Subgroup narrative comparisons were not carried out given the limited number of studies included and heterogeneity of the evidence.

### 3. Results

### 3.1. Selection of Studies

   As shown in Figure 1, we retrieved a total of 3565 records from the online databases, as well as 47 records identified manually in the reference lists of included studies. We identified 1023 duplicates, which were removed, leaving us with 2589 unique records to screen in stage 1. Of these, 2471 abstracts were excluded at stage 1, whereas 118 records moved to stage 2. The full-text screening resulted in 108 records being excluded either because the full-text was not available ($n = 4$) or not in English language ($n = 8$) or because

the study did not meet one of our population, exposure, or outcome criteria (*n* = 96). No studies were excluded based on study design. A total of 10 papers were included in the final review [36,40–48].

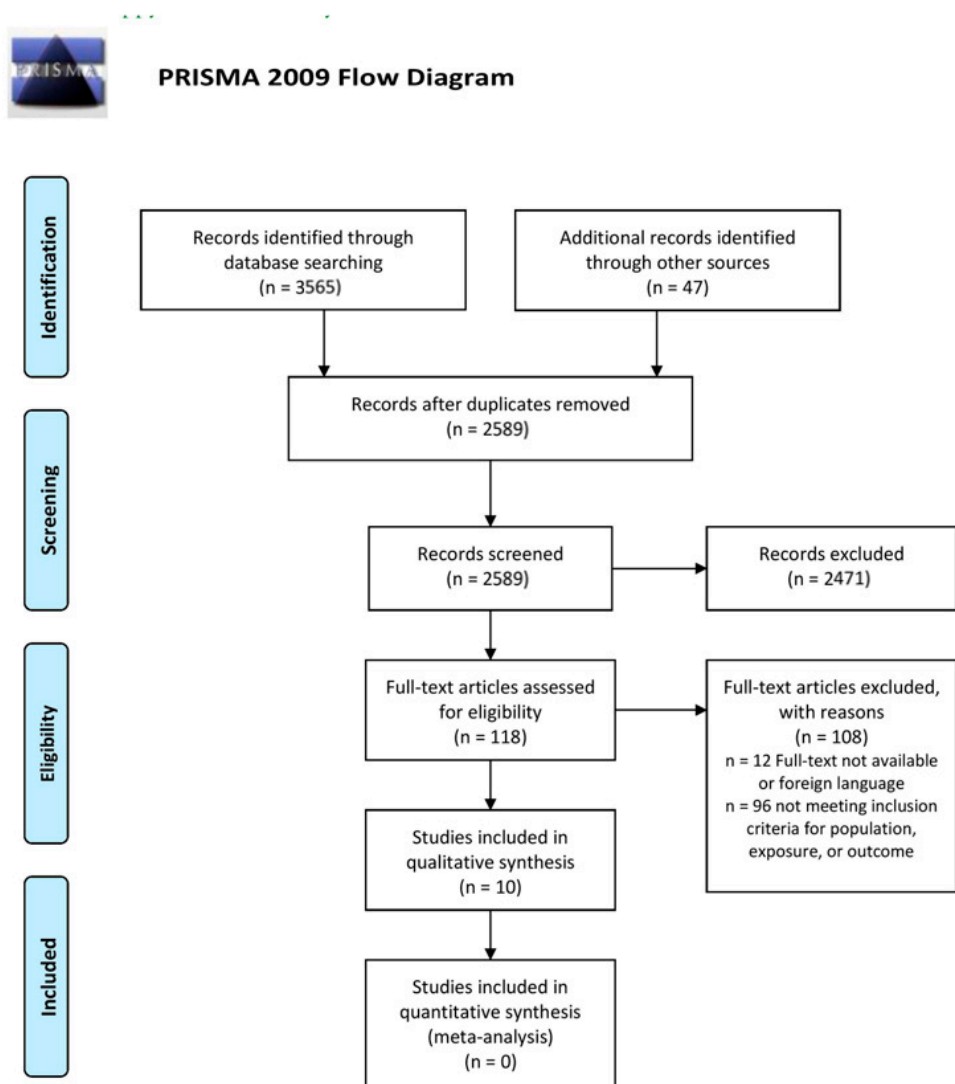

**Figure 1.** PRISMA flow chart of included studies [37].

### 3.2. Study Characteristics

Characteristics of the 10 included studies are presented in Supplementary File S3. In terms of country distribution, four out of 10 studies were conducted in the UK, of which two were in England [42,44], and two in Scotland [43,48]); two studies were in the USA [36,47]; and the remaining four studies were carried out respectively in Spain [46], Norway [41], Turkey [40], and New Zealand [45]. Study methodologies ranged from mixed methods [41,45,48], to qualitative [42,44,46], or quantitative surveys carried out either longitudinally [43,47] or cross-sectionally after the intervention [40]. One paper [36] presented a qualitative review of existing case studies.

Measures of psychological wellbeing varied greatly across studies (see Table 1 below), with none of the studies explicitly mentioning happiness and only three studies assessing wellbeing-related dimensions through standardised measures [43,47,48]. The remaining studies measured sense of belonging (for instance, through relational wellbeing or social capital) and/or agency (residential satisfaction, perceived safety, or quality), mostly with ad-hoc measures.

**Table 1.** Classification of included studies based on intervention approaches.

| Intervention Approach | Intervention Type | Outcome(s) | Study |
|---|---|---|---|
| Comprehensive | Neighbourhood regeneration via development of housing and roads | Belonging and agency–Residential satisfaction (ad-hoc measure) | [40] |
| | Community regeneration through housing redevelopment with refurbishment or demolition of buildings; refurbishing of public spaces | Belonging and agency (perceived social exclusion, ad-hoc measure) | [44] |
| | Ecological urban renewal: community-designed street murals, public benches, planter boxes, information kiosks with bulletin boards, trellises for hanging gardens, all positioned in the public right-of-way. | Depression (CESD-11), Wellbeing (SF-36), Social capital as measure of belonging (ad-hoc measure) | [47] |
| | Urban regeneration projects including creation of community gardens and bike paths and the redevelopment of brownfields, as well as housing revitilisation | Belonging and agency (unclear definition) | [36] |
| | Neighbourhood regeneration via development of housing and roads | Belonging and agency–Residential satisfaction (ad-hoc measure) | [40] |
| Focused | Green spaces | Agency-Satisfaction with the availability and 'quality of neighbourhood resources' | [41] * |
| | | Belonging-Relational wellbeing (behavioural observation) | [46] |
| | | Mental wellbeing (Warwick-Edinburgh Mental Wellbeing Scale) Stress (Perceived Stress Scale) Agency-Satisfaction with neighbourhood quality of life (ad hoc measure). | [48] |
| | Transport infrastructure | Agency-Satisfaction with the availability and 'quality of neighbourhood resources' | [41] * |
| | | Mental wellbeing: Warwick-Edinburgh Mental Well-being Scale; SF-8 | [43] |
| | | Agency-Residents' satisfaction. | [45] |
| | Security | Agency-Perceived safety | [42] |

Notes. * Discussed here in terms of impact on transport.

In terms of populations, six of the studies involved people with low income or living in socioeconomically deprived areas [36,41,43,45,47,48], although some studies looked at more than one category of vulnerable group. Three studies presented findings related to children or families with children [40,45,46], Three involved women [42,44,48], and one study considered older people [40]. None of the studies described interventions involving individuals with disabilities, race minorities, or migrants.

*3.3. Intervention Approaches*

In line with a previous systematic review on the mental health impact of urban design interventions [22], we categorised interventions based on their scope (see Table 1), as (a) having a comprehensive approach centred around urban regeneration or renewal [36,40,44,47]; or (b) employing a focused approach [41–43,45,46,48]. The latter group was further categorised based on the GAPS framework [23], in terms of changes to urban green space, transport infrastructure, or safety and security. A narrative synthesis of the intervention approaches will be presented in the following sections.

### 3.3.1. Interventions with a Comprehensive Approach

Three out of four studies in this category [40,44,47] explored the impact of multi-faceted urban regeneration projects on the wellbeing of different vulnerable groups, whereas one article [36] presented a critical review of two successful urban design initiatives oriented towards community development. The regeneration projects consisted mainly of housing redevelopment [36,40,44] and renewal of public spaces (in particular, [47]).

Two studies [40,44] reported a negative impact on wellbeing of housing/public space redevelopment on residents. Afacan [40] used a quantitative cross-sectional survey to understand residents' satisfaction and experiences associated with the redevelopment of housing and roads of a neighbourhood in Ankara (Turkey), focusing on older people and families with children. While the regeneration project improved overall residential satisfaction, older people reported higher perceptions of reduced road and social safety than younger residents, and families with children reported issues with accessibility. According to the author, the urban regeneration projects carried out in the neighbourhood led to the creation of low-quality urban spaces without elements of urban identity, causing issues around social development and displacement for the most vulnerable residents. Gosling [44] employed participant observation, focus groups, and interviews to explore women's experiences of social exclusion associated with the regeneration of a deprived area in a UK city; the regeneration process, mainly aimed at housing redevelopment and privatisation, was characterised by long delays with considerable loss of community during the process. Similar to Afacan [40], this study found reduced psychological wellbeing associated with the regeneration process, especially among single mothers and older women, due to feelings of powerlessness, concerns about being unable to return to their living area after the regeneration, and the decline in resident numbers and community groups.

Examples of successful interventions were given by Semenza et al. [47] and Pastor and Morello-Frosch [36]. In their survey study, Semenza et al. [47] explored an intervention in three low-to-moderate-income neighbourhoods of Portland, US, that aimed at engaging residents in the regeneration of their communities; this involved the introduction of street furniture (e.g., benches or bulletin boards), natural elements (e.g., trellises for hanging gardens) and street art (e.g., murals), all designed and executed by the community members. The initiative was associated with increased mental health, sense of community, social capital, and decreased depression.

In their analyses of community development initiatives for public health, Pastor and Morello-Frosch [36] described a group of initiatives aimed to empower communities for the promotion of positive health outcomes in 14 disadvantaged neighbourhoods across California, US. The programme focused on urban regeneration though community-led actions, including compact development, community gardens, walk and cycle infrastructure. While the authors do not provide a detailed evaluation of these initiatives, or measure their impact on mental wellbeing, they identified the use of community engagement strategies employed in these initiatives as the key to ensure that community regeneration projects had a tangible benefit on local resident, rather than displacing them.

The initiatives described by Semenza [47] and Pastor and Morello-Frosch [36] have in common an effort to develop grassroot community-led actions which nurtured community power and political engagement in socioeconomically disadvantaged areas. As noted by Pastor and Morello-Frosch [36], the active involvement of a diverse group of community members who had frequent interactions, not only helped to build trust between them, but also positively contributed to community development and civic engagement. Conversely, the regeneration initiatives described in Afacan [40] and Gosling [44] lacked a participatory element, employed a more top-down approach, and disrupted a sense of continuity for the community, which in turn led to low levels of perceived safety and residential satisfaction among women, low-income families with children, and older people. Overall, these studies point out that, when looking at urban interventions with a comprehensive approach, the ability to foster agency (i.e., participation in the process of change) and belonging (i.e.,

stability in sense of identity and community) in vulnerable groups through the process is key to determine a positive psychological and social impact.

3.3.2. Interventions with a Focused Approach

*Green spaces.* Two studies focused on urban parks and green spaces. Pérez del Pulgar et al. [46] employed archival and ethnographic observation methods to evaluate how the political/social production of green play spaces affected children's relational wellbeing (i.e., the experience of being well while socialising with others) in two case-study parks in Barcelona (Spain); these had been created as part of a long-term city regeneration process: One park presented diversity in design both within the park and in integration with the surrounding housing and road infrastructure, whereas the other was mainly characterised by unstructured green embedded in a gentrified area. It was found that diversity of design, which was influenced by the historical and social structures of the neighbourhood system, promoted free play and social interactions (as a proxy of relational wellbeing) more than unstructured green did. This was an indication, according to the authors, that the socio-material structures of neighbourhoods and residents' historical and social construction of the space were crucial in shaping the use of the park as a public and social space.

Ward Thompson et al. [48] used surveys and physiological methods to examine the impact of regenerated urban green spaces on quality of life and stress across two case studies in Scotland, UK, with a focus on disadvantaged communities and gender differences. They found that the mental wellbeing of individuals from deprived communities was mainly associated with the quality of the green space (e.g., perceived safety) rather than its quantity (i.e., amount of green space); this distinction was particularly relevant to women, who experienced higher levels of stress than men depending on social circumstances and frequency of use.

The role of quality of design for disadvantaged communities was also observed by Anthun et al. [41], who evaluated the enhancement of a green cycling and walking path along the coast in a suburban area in Norway, with the aim to improve the accessibility to the city and the quality of green spaces. By using questionnaire surveys, structured interviews, and digital counts), they found that the area residents were overall satisfied with the intervention and used the path, but residents with lower socioeconomic status were the least satisfied with the availability and quality of the new infrastructure, despite reporting high frequency of use. This result was ascribed to experienced issues with the location of the path as well as the perception that the path had been planned and implemented without inputs from all relevant community stakeholders, thus addressing the needs of some residents better than others.

*Transport.* Similar to Anthun, Macmillan et al. [45,49] presented a study aimed at assessing the effects of improved walking and cycling paths in low-income neighbourhoods in Auckland (New Zealand). The initiative employs a co-creative transdisciplinary approach, with the intervention designed collaboratively by researchers, the community, and transport investors. In a preliminary qualitative analysis of residents' perceptions of the new cycle path in one of the neighbourhoods involved, respondents highlighted a positive view on the path, but also indicated socioeconomic barriers and issues with accessibility and safety.

Considering transport infrastructure, Foley et al. [43] investigated the psychosocial effects of building a motorway through predominantly urban deprived neighbourhoods on local residents' wellbeing in Glasgow (UK) and found that, compared to residents of areas with existing motorways, people living nearer to the newly opened motorway experienced significantly reduced mental wellbeing over an eight-year period, particularly those with chronic conditions, who were the most negatively impacted in the long term. The negative impact was linked to increased traffic noise as well as reduced health-related quality of life, as a significant number of residents were not car owners, thus could not make use of the new road.

*Safety measures*. Lastly, Beebeejaun [42] described a qualitative participatory project focused on women's experiences of safety and security in six neighbourhoods of three UK cities that had all introduced technological monitoring systems (CCTV) in public spaces (Bristol, London, and Manchester). Women participating in the study reported how the introduced monitoring systems did not appear to increase a sense of safety, but rather gave a sense of policing and control, whereas design factors associated with natural surveillance on the street, such as lighting and landscaping, enhanced perceptions of safety and equal access when navigating public spaces.

Overall, although considering different groups (children, women, and low-income communities), studies on focused urban interventions revealed the complex social, cultural, and historical mechanisms that may influence psychological wellbeing in terms of access and perceived safety in green spaces. Across nature, transport or security interventions, the wellbeing derived from accessibility was associated less with the quantity of space available than with the perceived quality and safety of the transformation, particularly for women and low-income communities, as well as the ability to cater for sustainable living. Across interventions, the promotion/lack of residents' participation and empowerment emerged once again as important to the experience of change.

### 3.4. Risk of Bias

The quality assessment of included studies, carried out using the MMAT tool, is presented in Supplementary File S4. Eight of the 10 studies met the criteria to be considered empirical studies, while two of the included papers [36,45] could not be fully assessed as empirical studies because they presented as a review and a study design paper, respectively. Of the eight empirical studies included, six [40,41,43,44,46,48] met all the criteria for low risk of bias; however, two studies showed lack of clarity in terms of sample representativeness and/or adequacy of measurement [42,47].

### 4. Discussion

We identified 10 papers which met our inclusion criteria for urban design interventions or transformations with an impact on the psychological wellbeing of vulnerable groups. Overall, we found mixed results, in that only some of the interventions (comprehensive or focused green and transport infrastructure) were associated with increased residential satisfaction, quality of life, and wellbeing for low-income residents and women. This weak evidence is partially in line with what was found in previous investigations of urban regeneration and mental health [20–22]. In their systematic review of controlled studies, for instance, Moore et al. [22] identified small improvements in quality of life or social integration associated with green infrastructure interventions but no evidence of benefits associated with large regeneration projects. Like Moore [22], we observed considerable methodological heterogeneity, which limit our ability to draw firm conclusions on effectiveness overall or by intervention approach (comprehensive vs. focused). However, when we considered the experiences of vulnerable groups across studies, three crucial factors emerged as determinants of a positive effect of the intervention, independent of its scope, namely, participation, sustainability, and quality of design.

"Participation" here is intended as the ability of vulnerable groups to actively engage in the process of urban changes, thus having their needs appropriately represented. Grassroot, community-led interventions described in some of the studies, whether comprehensive or focused, appeared to be successful in empowering a sense of both agency (i.e., ability to access and use) and belonging (i.e., social stewardship and safety) among the most vulnerable residents, particularly if coming from a low socioeconomic background. Indeed, the lack of participatory planning and design was a key factor associated with low satisfaction rates and poor mental health outcomes among low-income groups, families with children, older people, and women. This resonates with previous literature highlighting the crucial role of participatory approaches in advancing a more equitable agenda for

urban regeneration [33,50] as well as developing a sense of transformative agency among the most marginalised citizens [51].

Another enabler of psychosocial wellbeing was the sense of continuity and sustainability experienced by vulnerable groups during and after the intervention; here, with "sustainability" we intend for the opportunities afforded to residents to sustain usual activities and their sense of identity to be both along a temporal dimension (e.g., the ability to remain in the area during and/or after the intervention), and from a spatial perspective (i.e., a sense of access and permeability across the community, beyond the home), particularly among low-income groups and women with regards to transport, green, and security measures. The benefits of gender-sensitive urban planning are supported in the literature, especially with regards to wellbeing derived from perceived safety [52]. Continuity was also observed as a factor for mental health in O'Campo et al. [32], who used conceptual mapping to highlight the importance of maintaining access to local amenities and services for positive mental wellbeing, especially in the most disadvantaged neighbourhoods. Similarly, another study [53] found that "access" is experienced differently by groups of low or high socioeconomic status based on land use and transport design. The need for spatial and temporal continuity highlighted in some of the included studies also suggests the relevance of using a systems approach to ensure a sustainable urban change, as discussed by Pérez del Pulgar [46], which is in line with growing calls for an ecological and systems-based approach to urban development [54]. Thus, sustainable urban changes should consider the complex relationships between urban design and wellbeing at both a spatial, social, and cultural level.

Linked to participation and sustainability, the reviewed studies, particularly those describing focused interventions, also highlighted the importance of the "quality" of certain features of urban environments, along with their quantity and availability, especially for lower-income groups and women who may have fewer personal resources to adapt to changes to their local green and open spaces [48]. Perceived quality associated with the spatial and social characteristics of new urban spaces was intended in the included studies as the opportunity to access the space easily and safely; in this sense, quality was a key determinant of experienced access in the included studies, and this is in line with what was highlighted in a recent systematic review of green spaces and mental health [55]. From a gender perspective, our findings are in line with previous research, which found that modifying the function of an urban space (i.e., designing for activity) led to higher ratings of perceived safety among women compared to interventions focused on vegetation or sanitation [56]. Perceived quality may also depend on the social circumstances of users of new spaces (e.g., low-income groups may need new transport or green infrastructure for instrumental rather than leisure purposes) [57], thus complicating the relationship between perceived space and wellbeing. This suggests that, while urban planners and designers need to ensure adequate amounts of land are allocated to green spaces and cycling/walking infrastructure, their mere existence may not necessarily lead to significant improvements on the mental wellbeing of vulnerable groups, particularly when these spaces do not consider the required qualities [58,59].

The three factors, participation, sustainability, and quality, associated with the positive psychological impact of urban interventions for vulnerable groups, support the argument within the GAPS framework [23] that designing urban spaces which promote activity, social empowerment, contact with nature and sense of safety, may be beneficial for different sociodemographic groups. More broadly, our findings support the pivotal role of "place-making" and its psychological dimensions for transformative people-centred urban planning [60,61], as well as the need to develop an interdisciplinary and holistic approach to urban transformations [13,62], in order to cater for both physical and mental wellbeing in a sustainable way [4], especially for the most vulnerable individuals from a socioeconomic and gender dimension.

These insights lend further support to the development of a sustainable urban agenda [6], for both healthy, happy, and equitable cities; however, our review has identified important

evidence gaps. The diversity in definitions of "psychological wellbeing" across the included studies begs the question as to what wellbeing, or indeed happiness, actually is, as highlighted by Moore et al. [22] in their systematic review. It is possible that happiness is locally determined, depending on the spatial, social, and cultural mechanisms at play as well as the population of interest. While this can limit the ability to measure psychological wellbeing universally, adopting a broad definition of wellbeing along the dimensions of agency (i.e., what a space allows a person to achieve, or the eudaimonic dimension of happiness) and belonging (i.e., how a space becomes meaningful to a person's identity and affective states, or the hedonic dimension of happiness), as employed for instance in models of urban ageing [63], may serve well as an overarching framework to understand psychologically sensitive urban changes. We found no interventions evaluating the experiences of people with disabilities or racial minorities. Despite Age Friendly [28], Child Friendly [27], and universal design [29] initiatives are increasing globally, we found in our search that most empirical studies focus on social aspects, while the psychological impact of the physical environment requires further investigation. Thus, further investigation of spatial inequalities across different vulnerable groups is needed, especially in terms of experiences of change, to advance the narrative on equitable urban living. We identified a bias in geographical representation of interventions, which were mainly located in Western countries (particularly the UK and US). This calls for more empirical evidence from the Global South and developing countries, as the experiences of urban spatial inequalities may be different from those explored in this review [64]. In addition, none of the included studies considered smart or technological interventions; these have an increasingly crucial role in the development of sustainable urban environments [65], and it would be interesting to explore how technology may enhance the experience of urban transformations for vulnerable groups [66]. Lastly, the quality of the studies included in this review was mixed and study methods were very heterogenous, warranting further research employing mixed-methods to elicit the complexities of urban change experiences, whilst attempting to increase methodological robustness using natural experiments, controlled before–after designs, and comparisons across more and less vulnerable groups.

This is, to the best of our knowledge, the first systematic review to synthesise the evidence of the impact of urban design and planning interventions on the psychological wellbeing of vulnerable groups; our search, which was enriched by integrating interdisciplinary knowledge from the fields of psychology, urban design and planning. Our focus on subjective experiences and thus our broad search in terms of study methods and definition of psychological wellbeing enabled us to identify key determinants of positive change that may be applicable across comprehensive and focused urban interventions. Nonetheless, while the search terms related to urban transformations were carefully chosen based on the interdisciplinary expertise of the research team, we acknowledge the difficulty in capturing the wide scope of urban improvements that may be relevant. Furthermore, methodological and construct heterogeneity hindered comparability across studies in terms of effectiveness. In addition, geographical bias for the available evidence limits the generalisability of our findings. Lastly, this review was limited to empirical research to tease out effectiveness; however, a review of grey literature could be useful to identify further case studies that might shed light on this topic.

## 5. Conclusions

While existing studies have evaluated the effects of urban environments on mental wellbeing, this systematic review established the evidence on how changes within urban design and planning practices affect the psychological wellbeing of vulnerable groups, who are the most sensitive to environmental opportunities and challenges. We found some evidence that urban interventions, which are centred around participation, sustainability, and quality of design are associated with increased residential satisfaction and wellbeing, particularly among communities with socioeconomic disadvantage, women and, to a lesser extent, families with children. This finding emphasises the importance for policymakers and designers of promoting urban design changes that are holistically embedded within

sustainable community development strategies, considering multiple stakeholders and systems, and in particular, the psychological and health benefits of introducing green, active and safe infrastructure. However, we identified important gaps in terms of lack of evidence for individuals with disabilities, migrants, or groups from racial minorities. These, together with methodological heterogeneity and limited geographical diversity, warrant further investigation on the topic of equitable urban transformations that foster the creation of happy places for all.

**Supplementary Materials:** The following are available online at https://www.mdpi.com/article/10.3390/su13148086/s1, Supplementary File S1: PRISMA Checklist, Supplementary File S2: Inclusion Criteria and Searches, Supplementary File S3: Study Characteristics; Supplementary File S4: Quality assessment.

**Author Contributions:** Conceptualization, M.C., S.S., and S.B.; methodology, M.C., S.S., and S.B.; software, M.C.; formal analysis, M.C., S.S., and S.B.; investigation, M.C., S.S., and S.B.; resources, M.C., S.S., and S.B.; data curation, M.C., S.S., and S.B.; writing—original draft preparation, M.C., S.S., and S.B.; writing—review and editing, M.C., S.S., and S.B. All authors have read and agreed to the published version of the manuscript.

**Funding:** This research received no external funding.

**Institutional Review Board Statement:** Not applicable.

**Informed Consent Statement:** Not applicable.

**Data Availability Statement:** No primary data collected for this review.

**Conflicts of Interest:** The authors declare no conflict of interest.

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
