# Peer review of "Envisioning Happy Places for All: A Systematic Review of the Impact of Transformations in the Urban Environment on the Wellbeing of Vulnerable Groups"

_sustainability, doi:10.3390/su13148086_

Round 1
Reviewer 1 Report
This paper describes an interesting and valuable systematic review study and contains information that will be of interest to those looking at validity and potential of making changes to urban environments to benefit population wellbeing.
Introduction:
P1 L40 'Frameworks of urban “liveability” have demonstrated that the design of urban outdoor spaces can determine health inequalities' This text is far more assertive than the evidence. I think we can safely say that 'Frameworks of urban “liveability” have demonstrated that the design of urban outdoor spaces is a determinant of health inequalities', but I don't think we can go much further. Maybe a 'key' determinant, if the evidence supports that.
NB. There is quite a but of text using the terminologies of 'happy' places and happiness. This does not seem well supported in terms of concepts and discourses around psychological health, wellbeing and happiness. I think equating positive 'wellbeing' with 'happiness' is too strong. The role of resilience as a competent of 'wellbeing' is quite established, I can be sad or upset or angry (ie. not happy) but still be in an excellent state of overall wellbeing if I have the capacity to bounce back. I suggest that the literature around happiness would need unpacking a more for these sentences to carry weight.
Terms of 'Happy' places and happiness sit uncomfortably with the systematic review itself and there is a lack of a critical review of the literature around happiness concepts. 'Happy' appears mainly in the tittle, the introduction and conclusion - is this required at all?
P2 L74 'this view on wellbeing suggests that urban spaces can truly become “happy places” if they are equitable'. This sentence may need review - it that really a view? What do we meant here by the word 'truly', it seems to be doing all the heavy lifting.
Methodology:
P3 L108: 'This systematic review drew from expertise in Architecture, Urban Planning, and Psychology, to assess the evidence through a transdisciplinary lens and to compare 109 critically the effectiveness of comprehensive and focused urban interventions, so to clarify 110 which may bear larger benefits for mental wellbeing' - Good to see this approach.
P4 L187 Some kind of glitch maybe in the pdf build?
'reporting large datasets that are deposited in a publicly available database should specify where the data have been deposited and provide the relevant accession numbers. If the accession numbers have not yet been obtained at the time of submission, please state that they will be provided during review. They must be provided prior to publication. '
Looking at the search terms in the supplementary material - and I know as an 'open system', urban design searches are fraught with capturing the best studies using search terms - a short paragraph maybe in the limitations or problems of reducing the wide scope of urban improvements types to a series of search terms may be helpful. I was certainly surprised that there were only 10 studies meeting your requirements in this subject area. I notice terms such as improvement, redevelopment, housing and neighbourhood were not used in themselves.
Results:
P5 L195 Some kind of glitch maybe in the pdf build?
'This section may be divided by subheadings. It should provide a concise and precise 195 description of the experimental results, their interpretation, as well as the experimental 196 conclusions that can be drawn. '
Conclusion
Can we say anything for policy-makers and designers about better interventions themselves?
Author Response
This paper describes an interesting and valuable systematic review study and contains information that will be of interest to those looking at validity and potential of making changes to urban environments to benefit population wellbeing.
Response – Thank you for your positive feedback.
R1C1: Introduction: P1 L40 'Frameworks of urban “liveability” have demonstrated that the design of urban outdoor spaces can determine health inequalities' This text is far more assertive than the evidence. I think we can safely say that 'Frameworks of urban “liveability” have demonstrated that the design of urban outdoor spaces is a determinant of health inequalities', but I don't think we can go much further. Maybe a 'key' determinant, if the evidence supports that.
Response – We agree with the Reviewer. We have reworded the sentence (p.1 L47-48) to read: “increasingly suggest that the design of urban outdoor spaces can be a determinant of health inequalities”.
R1C2: NB. There is quite a but of text using the terminologies of 'happy' places and happiness. This does not seem well supported in terms of concepts and discourses around psychological health, wellbeing and happiness. I think equating positive 'wellbeing' with 'happiness' is too strong. The role of resilience as a competent of 'wellbeing' is quite established, I can be sad or upset or angry (ie. not happy) but still be in an excellent state of overall wellbeing if I have the capacity to bounce back. I suggest that the literature around happiness would need unpacking a more for these sentences to carry weight. Terms of 'Happy' places and happiness sit uncomfortably with the systematic review itself and there is a lack of a critical review of the literature around happiness concepts. 'Happy' appears mainly in the tittle, the introduction and conclusion - is this required at all?
Response – We thank the Reviewer for this point. Indeed, happiness and subjective wellbeing can be defined in multiple ways; to this end, we have now clarified in the Introduction our choice of definition of happiness and psychological wellbeing, as follows (pp.1-2, lines 39-46):
“the Global Happiness Policy Report 2018 [7] considers urban planning as one of the key determinants of happiness. In this review, in line with Dodge et al. [8] we consider happiness in the broad sense of a condition of positive and sustained subjective wellbeing [9]encompassing both the hedonic dimension of positive affective states [10] and the eudaimonic dimension of positive psychological functioning, flourishing and development (see for a discussion [11]). To this end, and in line with previous research on the topic [11,12], the terms happiness and psychological wellbeing are used interchangeably hereafter. ”
We have also expanded the literature review section to better highlight the literature we draw upon to study the link between urban transformations and happiness in its broad meaning of subjective psychological wellbeing (p2 L53-57):
“Since the publication of Montgomery’s “Happy City” book [19], there have been increasing attempts to develop ways to measure the link between urban development and happiness [12]; however, there is mixed evidence on the association between urban transformation and psychological wellbeing ”
We have clarified how we link a sense of belonging and/or agency determined by urban spaces to our definition of happiness at p.2 L70-71 and L82-83:
“Thus, urban spaces that are created to foster a sense agency (or a positive eudaimonic state) and belonging (i.e., positive hedonic or affective state) may promote happy citizens”
In line with this, we have rephrased the Outcome inclusion criterion section (p.4 L.154-156) to clarify our choice of criteria:
“Given the interchangeability with which happiness and psychological wellbeing are used in the literature, as discussed in the introduction, we adopted here a broad definition [8] to encompass both direct and indirect measures of “being psychologically well” in the community”
We agree that the concepts of happiness or subjective wellbeing are not clearly defined in the literature, and we note this in the Discussion section, which we have now amended to better clarify how we see an urban space contributing to happiness (p.11 L473-483, the yellow highlights indicate the revised parts):
“The diversity in definitions of “psychological wellbeing” across the included studies begs the question on what wellbeing, or indeed happiness, actually is, as highlighted by Moore et al [22] in their systematic review. It is possible that happiness is locally determined, depending on the spatial, social, and cultural mechanisms at play as well as the population of interest. While this can limit the ability to measure psychological wellbeing universally, adopting a broad definition of wellbeing along the dimensions of agency (i.e., what a space allows a person to achieve, or the eudaimonic dimension of happiness) and belonging (i.e., how a space becomes meaningful to a person’s identity and affective states, or the hedonic dimension of happiness), as employed for instance in models of urban ageing [63], may serve well as an overarching framework to understand psychologically sensitive urban changes.”
We hope that the abovementioned changes have clarified our choice of term “happiness”. Lastly, as we note in the Results section (p.7 L237) none of the included studies employed a standardised measure of happiness but considered measures of psychological wellbeing that fall within a broad definition of happiness. We feel that excluding studies based on a strict measure of happiness would have considerably limited our ability to conduct this review.
R1C3: P2 L74 'this view on wellbeing suggests that urban spaces can truly become “happy places” if they are equitable'. This sentence may need review - it that really a view? What do we meant here by the word 'truly', it seems to be doing all the heavy lifting.
Response – We acknowledge the semantic confusion here. We have rephrased the sentence to enhance clarity, as follows (p.2 L86-89):
“Applied to urban design, this view on psychological wellbeing suggests that equity of urban spaces is crucial to create “happy places”, i.e., places that support the psychological needs of those individuals who are the most vulnerable.”
R1C4: Methodology:P3 L108: 'This systematic review drew from expertise in Architecture, Urban Planning, and Psychology, to assess the evidence through a transdisciplinary lens and to compare critically the effectiveness of comprehensive and focused urban interventions, so to clarify which may bear larger benefits for mental wellbeing' - Good to see this approach.
Response - We thank the Reviewer for the positive feedback on this aspect.
R1C5 - P4 L187 Some kind of glitch maybe in the pdf build? 'reporting large datasets that are deposited in a publicly available database should specify where the data have been deposited and provide the relevant accession numbers. If the accession numbers have not yet been obtained at the time of submission, please state that they will be provided during review. They must be provided prior to publication. '
Response – Thank you for identifying this error. This is a part of the submission template that was inadvertently left in the manuscript. That has been removed.
R1C6 - Looking at the search terms in the supplementary material - and I know as an 'open system', urban design searches are fraught with capturing the best studies using search terms - a short paragraph maybe in the limitations or problems of reducing the wide scope of urban improvements types to a series of search terms may be helpful. I was certainly surprised that there were only 10 studies meeting your requirements in this subject area. I notice terms such as improvement, redevelopment, housing and neighbourhood were not used in themselves.
Response – Thank you for this point. Indeed, the research team spent quite a significant amount of time discussing and optimising the search terms. We agree with the Reviewer that the scope is quite broad, and we have acknowledged this in the Discussion (p.13 L.509-512):
“While the search terms related to urban transformations were carefully chosen based on the interdisciplinary expertise of the research team, we acknowledge the difficulty in capturing the wide scope of urban improvements that may be relevant.”
Based on the expertise of the research team, we opted for terms that would be broad enough to potentially capture the terms highlighted by the Reviewer (e.g., neighbourhood) while keeping the focus on transformations and changes, rather than urban design as a status quo). We feel that our search result of over 3,500 records supports the validity of choice. We have now made a clearer point about the strategies we employed to reduce error in our selection process (p.4 L184-186):
“The piloting stage and involvement of all authors in the screening process were adopted to limit potential bias or error in study selection.”
While agree with the Reviewer that a small number of studies has been included in the review, this is mainly due to the focus on transformations and changes in urban design rather than looking at the status quo, which has been researched already in existing systematic reviews. Although this focus narrows down the inclusion criteria considerably, it was felt that this would enable us to better address our specific study objectives.
R1C7 - Results: P5 L195 Some kind of glitch maybe in the pdf build? 'This section may be divided by subheadings. It should provide a concise and precise 195 description of the experimental results, their interpretation, as well as the experimental 196 conclusions that can be drawn. '
Response – Apologies once again for this error. The template text has been removed.
R1C8 - Conclusion - Can we say anything for policy-makers and designers about better interventions themselves?
Response – We have expanded the Conclusion section to better highlight suggestions for urban interventions, as follows (p.13 L526-530):
“This finding emphasises the importance for policymakers and designers of promoting urban design changes that are holistically embedded within sustainable community development strategies, considering multiple stakeholders and systems, and in particular the psychological and health benefits of introducing green, active and safe infrastructure.”
Reviewer 2 Report
This is a systematic review that followed PRISMA guidelines and pre-registered with PROSPERO. The authors attempted to review previous studies relating to whether and how urban planning and design interventions impact the wellbeing of vulnerable groups. The topic is an important topic of understanding how urban planning makes a "happy place".
The eligibility criteria need further clarification. For the double criterion under the criteria "intervention", I assume the authors were saying that both conditions need to be fulfilled in order to be selected? If so, please state clearly instead of using the ambiguous term "double criteria". The authors may want to rephrase to something much more straightforward, e.g., studies need to involve both change in planning/design of the urban environment and in an urban area (as opposed to rural..) to be included in this study.
Also, please state clearly whether all 5 criteria listed from lines 129-147 need to be fulfilled in order to be selected. Please also list clearly what is under "or" condition and what is under "and" condition. For example, for study design, I assume it should be "observational studies, quantitative OR qualitative, controlled studies, OR case studies"?
The authors did not list their search terms. What were the terms used in searching in the first stage? (section 2.3-2.4) All these details need to be carefully listed so that other researchers can take the method section and reproduce the same findings.
I am a bit surprised that the authors only ended up with 10 studies. This is very small for a systematic review. This is particularly problematic because the authors excluded some studies because they could not find the full text. Did they attempt to contact the authors? If so, please also state it in the manuscript to justify the small sample.
The authors conducted a quality assessment of the included studies using the MMAT tool included in supp. file 4. The tables were fine, but how did the authors draw conclusion that "overall, risk of bias was low to moderate"? Did they use a certain standard to conclude that their risk was low? If so, please state clearly in the manuscript.
Author Response
We thank the Reviewer for the positive and constructive feedback. Please find our responses below.
This is a systematic review that followed PRISMA guidelines and pre-registered with PROSPERO. The authors attempted to review previous studies relating to whether and how urban planning and design interventions impact the wellbeing of vulnerable groups. The topic is an important topic of understanding how urban planning makes a "happy place".
R2C1: The eligibility criteria need further clarification. For the double criterion under the criteria "intervention", I assume the authors were saying that both conditions need to be fulfilled in order to be selected? If so, please state clearly instead of using the ambiguous term "double criteria". The authors may want to rephrase to something much more straightforward, e.g., studies need to involve both change in planning/design of the urban environment and in an urban area (as opposed to rural..) to be included in this study.
Response – We thank the Reviewer for this point. We have rephrased the Intervention inclusion criterion to enhance readability, as follows (pp.3-4 L145-150):
“Intervention (exposure): Studies were included if they described a) one or more urban interventions (i.e., change in planning or design of the urban environment as opposed to describing the status quo of urban design), AND b) one or more interventions occurring in urban area as opposed to rural, natural, or wild areas.”
We have also updated Supplementary file 2 to clarify this point.
R2C2 - Also, please state clearly whether all 5 criteria listed from lines 129-147 need to be fulfilled in order to be selected. Please also list clearly what is under "or" condition and what is under "and" condition. For example, for study design, I assume it should be "observational studies, quantitative OR qualitative, controlled studies, OR case studies"?
Response – Thank you. We have clarified at the end of section 2.2 (p.4 lines 163-165) that “In line with PICOS guidelines, all the above five criteria needed to be met for a study to be included in the review. Further details on the logic strategy used to select studies are presented in Supplementary File 2.”
We have also amended the study design criterion to clarify that we would include observational studies that were either quantitative OR qualitative (p.4 L161).
R2C3 - The authors did not list their search terms. What were the terms used in searching in the first stage? (section 2.3-2.4) All these details need to be carefully listed so that other researchers can take the method section and reproduce the same findings.
Response – Apologies if this has caused confusion. All search terms and search string are included in Supplementary File 2, which has been submitted with the other supplementary files in a compressed folder. We have now made a clear reference to this in section 2.3 L167
R2C4 - I am a bit surprised that the authors only ended up with 10 studies. This is very small for a systematic review. This is particularly problematic because the authors excluded some studies because they could not find the full text. Did they attempt to contact the authors? If so, please also state it in the manuscript to justify the small sample.
Response – While agree with the Reviewer that a small number of studies has been included in the review, this is mainly due to the focus on transformations and changes in urban design rather than looking at the status quo, which has been researched already in existing systematic reviews. Although this focus narrows down the inclusion criteria considerably, it was felt that this would enable us to better address our specific study objectives. We have made our focus more evident in the Discussion section, p.10 L395.
We have now clarified in section 3.1 (L221) that there were 4 instances of records for which we could not access the full-text and 8 records for which we could not retrieve a full-text in English language We did attempt to contact paper authors of records for which we could not access a full-text, but without success, with authors either not responding (in most cases) or indicating that they had no access to a copy of full-text to share (one instance).
We have clarified in section 2.3 (L171-173) that: “Where the full-text of an abstract could not be accessed by the research team an attempt was made to contact the study authors via email.”
R2C5 - The authors conducted a quality assessment of the included studies using the MMAT tool included in supp. file 4. The tables were fine, but how did the authors draw conclusion that "overall, risk of bias was low to moderate"? Did they use a certain standard to conclude that their risk was low? If so, please state clearly in the manuscript.
Response – We thank the Reviewer for this point and acknowledge that that sentence is too generic. We have amended section 3.4 (p.10 L383-392) to read as follows:
“The quality assessment of included studies, carried out using the MMAT tool, is presented in Supplementary File 4. Eight of the 10 studies met the criteria to be considered empirical studies, while two of the included papers [36,45] could not be fully assessed as empirical studies because presented respectively as a review and a study design. Of the eight empirical studies included, six [40,41,43,44,46,48] met all the criteria for low risk of bias; however, two studies showed lack of clarity in terms of sample representativeness and/or adequacy of measurement [42,47].”